# SCALE: Synergized Collaboration of Asymmetric Language Translation Engines

## Abstract

In this paper, we introduce SCALE, a collaborative framework that connects compact Specialized Translation Models (STMs) and general-purpose Large Language Models (LLMs) as one unified translation engine. By introducing translation from STM into the triplet in-context demonstrations, SCALE unlocks refinement and pivoting ability of LLM, thus mitigating language bias of LLM and parallel data bias of STM, enhancing LLM speciality without sacrificing generality, and facilitating continual learning without expensive LLM fine-tuning. Our comprehensive experiments show that SCALE significantly outperforms both few-shot LLMs (GPT-4) and specialized models (NLLB) in challenging low-resource settings. Moreover, in Xhosa to English translation, SCALE experiences consistent improvement by a 4 BLEURT score without tuning LLM and surpasses few-shot GPT-4 by 2.5 COMET score and 3.8 BLEURT score when equipped with a compact model consisting of merely 600M parameters. SCALE could also effectively exploit the existing language bias of LLMs by using an English-centric STM as a pivot for translation between any language pairs, outperforming few-shot GPT-4 by an average of 6 COMET points across eight translation directions. Furthermore we provide an in-depth analysis of SCALE's robustness, translation characteristics, and latency costs, providing solid foundation for future studies exploring the potential synergy between LLMs and more specialized translation models.

## 1 Introduction

Large Language Models (LLMs) have recently revolutionized the field of natural language processing (OpenAI, 2023; Touvron et al., 2023; Peng et al., 2023), significantly influencing machine translation (MT) by delivering exceptional performance without requiring a bilingual corpus, particularly in high-resource languages (Brown et al., 2020; Garcia et al., 2023). Moreover, as a unified multi-task learner, LLMs represent a substantial step towards artificial general intelligence (Bubeck et al., 2023), with the potential to transcend not only the language barriers emphasized by previous MT research but also cultural boundaries through a simple "translate and explain" prompt.

Despite their advancements, LLM-based translation systems still confront several challenges. Firstly, there exists a significant language bias towards English (e.g., 92.1% of the GPT-3 pre-training corpus is English, while French, the second largest, represents only 1.8%), which significantly constraints multilingual translation performance, especially for those low-resource languages (Scao et al., 2022; Hendy et al., 2023). Secondly, as a highly effective approach to improve system performance, fine-tuning (Hu et al., 2021; Dettmers et al., 2023) is non-trivial for LLMs due to (1) the trade-off between speciality and generality (Lin et al., 2023; Cheng et al., 2023a), and (2) the prohibitively high cost associated with tuning large-scale models. In contrast, traditional Specialized Translation Models (STMs)—those based on encoder-decoder architecture, trained with supervision and significantly smaller in size (Sutskever et al., 2014; Vaswani et al., 2017)—serve as specialists for specific translation tasks and could be efficiently fine-tuned. Nevertheless, these models exhibit limitations in general language capabilities and may be prone to parallel data bias, such as the memorization of low-quality samples (Raunak et al., 2022), due to restricted model capacity.

In this paper, we demonstrate for the first time the possibility to unify these two asymmetric translation engines in a single framework. Our work, SCALE, connects LLMs and STMS by utilizing the LLM's most enigmatic capability: in-context learning. Rather than employing source-target pairs as

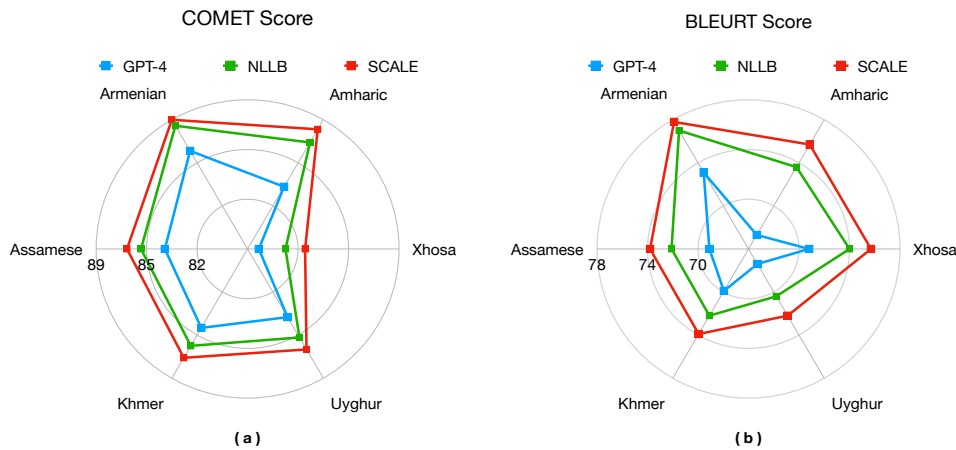

Figure 1: Translation results of few-shot LLM (GPT-4), STM (NLLB) and SCALE (ours) for six low-resource languages measured by COMET and BLEURT.

in conventional few-shot translation (Garcia et al., 2023; Vilar et al., 2023), SCALE would first sample translations from a STM and then use triplets consisting of a source sentence, an STM-generated set and a target sentence as in-context demonstrations to unlock the refinement and pivoting ability of LLMs. With SCALE, we could (1) mitigate both language bias of LLMs by utilizing an STM that concentrates on a specific language pair, and parallel data bias of STMs by using a general-purpose LLM as the main body of the system; (2) enhance the speciality of LLMs without compromising generality; (3) facilitate continual learning within the framework by updating only the lightweight STM, thus avoiding expensive LLM fine-tuning. By employing SCALE, we create a more efficient and effective system that combines the best of both translation engines.

Our comprehensive experiments reveal that SCALE considerably outperforms LLMs (e.g., GPT-4) and STMs (NLLB Team et al., 2022) in the challenging low-resource setting, as depicted in Figure 1. Moreover, in Xhosa→English translation, SCALE experiences consistent improvement by a 4 BLEURT score without tuning LLM and surpasses few-shot GPT-4 by 2.5 COMET score and 3.8 BLEURT score when equipped with a compact model consisting of merely 600M parameters. Remarkably, SCALE can effectively exploit the existing language bias of LLMs by using an English-centric STM as a pivot for translation between any language pairs, outperforming few-shot GPT-4 by an average of 6 COMET points across eight translation directions. Furthermore, we conduct an in-depth analysis of the robustness, translation characteristics, and latency costs associated with SCALE. Our findings provide valuable insights and encourage further research in this field.

## 2 THE SCALE FRAMEWORK

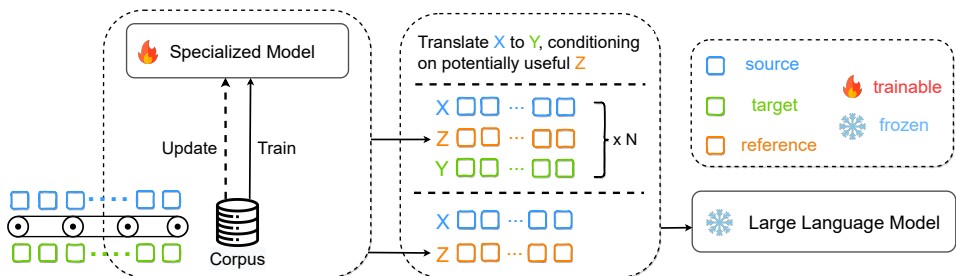

Figure 2: The SCALE framework, comprised of a lightweight specialized model and a frozen large language model with triplet in-context demonstrations.

In this section, we present the proposed SCALE method and provide an overview illustrated in Figure 2. Popularized by GPT-3 (Brown et al., 2020), In-context Learning (ICL) allows LLMs to

perform a wide variety of tasks, even newly created ones (Bills et al., 2023), by leveraging few-shot learning with a limited number of demonstrations. For a translation task from a source language $\mathcal{X}$ to a target language $\mathcal{Y}$, an LLM with parameters $\theta$ carries out ICL by conditioning on $k$ source-target paired examples $\mathbb{E} = (x_1, y_1) \oplus (x_2, y_2) \oplus ...(x_k, y_k)$ and the test source sentence $x$, generating the target $y$ in an auto-regressive manner as $y_t \sim p_\theta(y_t | \mathbb{E}, x, y_{<t})$. In this scenario, the LLM must analyze the provided examples to discern the input distribution, output distribution, input-output mapping, and formatting to successfully complete the task (Press et al., 2022; Wei et al., 2023). Different from conventional ICL, SCALE introduces an intermediate variable $\mathbb{Z}$ as reference between source $x$ and target $y$, transforming each demonstration example into a triplet $(x, \mathbb{Z}, y)$. The variable $\mathbb{Z}$ is a generation set sampled from a specialized translation model $\mathbf{M}_{\mathcal{X} \mapsto \mathcal{Y}}$ trained on a labeled dataset. The final input to the LLM consists of the instruction, demonstrations, and source sentence combined in a prompt template: $\mathcal{T}((x_1, \mathbb{Z}_1, y_1) \oplus (x_2, \mathbb{Z}_2, y_2)... \oplus (x_k, \mathbb{Z}_k, y_k)), (x, \mathbb{Z}))$. Unlike language understanding tasks that have fixed label set (Xu et al., 2023), the hypothesis space of translation model is actually infinite, so we could sample multiple generation paths from STM for one single source sentence to provide a more comprehensive generation guide for LLM. The SCALE framework, though conceptually straightforward, demonstrates several advantages over STMs and LLMs, as highlighted below:

**Refinement** For $\mathcal{X}$ to $\mathcal{Y}$ translation task, when the intermediate variable $\mathbb{Z}$ is from $\mathbf{M}_{\mathcal{X} \mapsto \mathcal{Y}}(x)$, SCALE essentially conduct few-shot learning in a multi-task way by introducing an additional refinement task. Refinement has long been proved effective in MT (Xia et al., 2017; Cheng et al., 2022). And this also holds true for LLM-based translation. In this refinement process, we pass sampled sentences and their confidence score (probability score) from STM to an LLM. The LLM then digests the information carried by the sampled set and infers the generation space of the STM, which guides the LLM to generate the output that is more consistent with the local data distribution (Xu et al., 2023). And since the final translation is delivered by an LLM, SCALE could also mitigate the parallel data bias from STMs and exhibit robustness by not merely copying and pasting the draft translation from STMs as shown in §5.3.

**Pivoting** Considering the predominantly English-centric nature of most LLMs (Brown et al., 2020; Touvron et al., 2023), SCALE could employ an intermediate variable $\mathbb{Z}$ from $\mathbf{M}_{\mathcal{X} \mapsto \text{English}}(x)$ where the target language $\mathcal{Y}$ is not necessarily English. And here $\mathbb{Z}$ serves as a pivot point for LLMs to enhance their understanding of the source sentence and yield improved translations. This can also be regarded as a form of knowledge transfer from high-resource languages to low-resource languages (Chen et al., 2017; Kim et al., 2019; Jiao et al., 2023).

**Updating** A significant limitation of the existing LLM-based translation systems is the inherent complexity of LLM continual learning. This complexity arises from several factors, including the delicate balance between speciality and generality (Lin et al., 2023), the catastrophic forgetting problem (Yong et al., 2023), and the substantial computational demands (Dettmers et al., 2023). In contrast, the SCALE framework offers a more efficient and streamlined approach to continuous updating. By exclusively and effectively updating the lightweight $\mathbf{M}_{\mathcal{X} \mapsto \cdot}$ component, the framework ensures that the LLM remains untouched, thus preserving its general language capabilities. This selective updating process not only mitigates the issue of catastrophic forgetting but also reduces the computational burden of fine-tuning associated with LLM-based translation systems.

## 3 EXPERIMENTAL SETUP

### 3.1 DATASET

Our evaluation datasets encompass a diverse set of languages, spanning both low- and high-resource settings and deriving from various language families. To facilitate reproducibility and data sharing, all our evaluation datasets come from the `devtest` split of Flores-200 (NLLB Team et al., 2022), a publicly available many-to-many evaluation data set covering 200 languages from all over the world.

### 3.2 TRANSLATION SYSTEMS

We compare our approach with cutting-edge academic systems including both specialized models and LLMs, as well as one commercial system, Microsoft Translator.

We have two strong specialized models:

- **M2M100** (Fan et al., 2021) is the first multilingual encoder-decoder translation model that can translate between any pair of 100 languages without relying on English data.
- **NLLB** (NLLB Team et al., 2022) is a supervised translation model suite covering from 169M to 54.5B (MOE) parameters with encoder-decoder architecture and capable of delivering high-quality translations directly between 200 languages.

For few-shot LLMs, we consider:

- **XGLM** (Lin et al., 2022) is a multilingual generative language models trained on a corpus covering a diverse set of languages and the largest XGLM-7.5B model outperforms comparable sized GPT-3 model in multilingual setting.
- **GPT-3.5** is a GPT model specially optimized for conversational purpose and shows remarkable performance in machine translation tasks (Jiao et al., 2023).
- **GPT-4** (OpenAI, 2023) is the latest and the most powerful version of GPT-series.

We use both GPT-3.5 and GPT-4 from Microsoft Azure OpenAI Service. Without further notice, the number of few-shot samples in LLM and SCALE are set to 10 and the sample selection strategy follows Agrawal et al. (2022). The prompt we use could be found in the Appendix A.1.

## 3.3 EVALUATION METRICS

Because neural metrics have shown higher correlation with human preference (Freitag et al., 2022; Rei et al., 2020) and are widely adopted by recent literatures (Hendy et al., 2023; Garcia et al., 2023), we mainly evaluate our system with (1) **COMET-22**, a reference-based neural metric (Rei et al., 2022a) combining direct assessments, sentence-level score, and word-level tags from multi-dimensional quality metrics error annotations, (2) **COMETKiwi**, a refrence-free quality estimation model from Rei et al. (2022b), and (3) **BLEURT** (Sellam et al., 2020), a learnable evaluation metric with a regression model trained on ratings data. For completeness, we also include the results of lexical metrics such as spBLEU (NLLB Team et al., 2022) and chrF++ (Popovic, 2017).

## 4 EXPERIMENTAL RESULTS

In this section, we conduct various experiments to show the effectiveness of our framework. In §4.1, we verify the effectiveness of the refinement ability within SCALE by comparing with STMs and few-shot LLMs. In §4.2, we focus on non-English pairs to test the pivoting ability of SCALE. In §4.3, we show the continual learning results of SCALE with a fixed LLM and an evolving STM.

## 4.1 SCALE REFINEMENT

To evaluate the refinement capabilities of SCALE, this section primarily concentrates on low-resource languages, which currently pose significant challenges for few-shot LLMs. Our approach showcases its versatility by incorporating languages from diverse families and scripts, including Assamese (asm_Beng), Armenian (hye_Armn), Amharic (amh_Ethi), Xhosa (xho_Latn), Uyghur (uig_Arab), Khmer (khm_Khmr), Nepali (npi_Deva), and Sindhi (snd_Arab). For additional data details, please refer to the Appendix A.2.

We adopt three kinds of baseline systems as described in §3.2. For supervised NLLB model suite, we choose the NLLB-3.3B version, and for SCALE-refine, the LLM is GPT-4 the STM is also NLLB-3.3B for fair comparison.

The results are displayed in Table 1. As observed, few-shot LLMs, including GPT-4, significantly trail behind specialized models in all translation directions. Even with Xhosa belonging to the same language family as English, the GPT-4 model fails to deliver comparable results to NLLB model. In contrast, our framework, by combining LLMs and STMs, demonstrates superior performance over few-shot GPT-4 by an averaged 2.96 COMET scores and 5 BLEURT scores, and surpasses the strong NLLB model in 8/8 directions. Interestingly, when the performance gap is substantial

| | COMET-22 | COMETKiwi | BLEURT | spBLEU | COMET-22 | COMETKiwi | BLEURT | spBLEU |
|---|---|---|---|---|---|---|---|---|
| | asm_Beng | | | | hye_Armn | | | |
| NLLB | 85.6 | 82.8 | 72.1 | 33.9 | 88.3 | 87.5 | 77.0 | 43.0 |
| M2M100 | n/a | n/a | n/a | n/a | 75.9 | 76.5 | 58.9 | 23.7 |
| Microsoft | 83.5 | 81.7 | 68.8 | 29.6 | 85.2 | 85.0 | 71.5 | 34.6 |
| XGLM | 62.7 | 57.8 | 38.8 | 3.7 | 43.9 | 50.2 | 20.5 | 0.2 |
| GPT-3.5 | 78.6 | 76.7 | 61.0 | 18.1 | 77.0 | 77.2 | 60.5 | 19.4 |
| GPT-4 | 83.9 | 80.9 | 69.1 | 27.9 | 86.2 | 86.0 | 73.1 | 35.6 |
| SCALE-refine | **86.6** | **83.2** | **73.8** | 34.1 | **88.8** | **88.0** | **77.8** | 42.3 |
| | amh_Ethi | | | | xho_Latn | | | |
| NLLB | 86.9 | 84.5 | 73.6 | 36.4 | 80.7 | 65.8 | 74.0 | 40.1 |
| M2M100 | 72.3 | 72.0 | 54.8 | 18.5 | 68.0 | 62.1 | 59.0 | 25.7 |
| Microsoft | 87.5 | 84.6 | 74.7 | 41.9 | n/a | n/a | n/a | n/a |
| XGLM | 50.2 | 43.9 | 17.8 | 0.1 | 39.6 | 41.7 | 37.1 | 1.6 |
| GPT-3.5 | 58.8 | 54.2 | 31.7 | 3.4 | 69.1 | 65.5 | 58.3 | 21.9 |
| GPT-4 | 83.2 | 81.9 | 67.3 | 27.1 | 78.8 | 67.1 | 70.8 | 34.5 |
| SCALE-refine | **88.0** | **85.3** | **75.7** | 37.6 | **82.1** | **67.3** | **75.7** | 40.0 |
| | uig_Arab | | | | khm_Khmr | | | |
| NLLB | 85.4 | 84.4 | 70.4 | 27.5 | 86.1 | 85.4 | 72.2 | 35.4 |
| M2M100 | n/a | n/a | n/a | n/a | 69.6 | 71.6 | 54.0 | 17.6 |
| Microsoft | 82.7 | 81.7 | 66.2 | 21.6 | 80.2 | 80.5 | 63.3 | 25.6 |
| XGLM | 37.1 | 52.8 | 16.9 | 0.2 | 48.6 | 53.7 | 21.6 | 0.7 |
| GPT-3.5 | 73.7 | 74.2 | 53.0 | 11.6 | 73.3 | 73.0 | 53.2 | 13.9 |
| GPT-4 | 83.7 | 82.8 | 67.4 | 23.1 | 84.6 | 84.0 | 69.9 | 29.1 |
| SCALE-refine | **86.4** | **85.0** | **72.2** | 27.9 | **87.1** | **85.9** | **73.9** | 34.7 |
| | npi_Deva | | | | snd_Arab | | | |
| NLLB | 90.4 | 88.3 | 77.1 | 45.0 | 86.9 | 79.5 | 75.5 | 44.4 |
| M2M100 | 75.2 | 73.6 | 55.1 | 21.2 | 49.8 | 47.2 | 39.2 | 6.4 |
| Microsoft | 89.8 | 88.2 | 75.3 | 42.8 | 83.6 | 77.4 | 70.4 | 38.5 |
| XGLM | 72.9 | 67.0 | 48.8 | 8.3 | 53.8 | 45.1 | 29.8 | 1.8 |
| GPT-3.5 | 87.2 | 85.4 | 69.9 | 29.3 | 75.6 | 68.1 | 58.8 | 17.3 |
| GPT-4 | 90.2 | 88.1 | 76.3 | 40.8 | 83.2 | 75.3 | 69.9 | 32.3 |
| SCALE-refine | **91.1** | **88.8** | **78.1** | 44.0 | **87.5** | **79.5** | **76.6** | 42.9 |

Table 1: Translation results of eight low-resource languages to English. The best results are in **bold** and the second best are with underscore. SCALE-refine is compared with specialized model (NLLB, M2M), commercial system (MS Translator) and few-shot LLM (XGLM, GPT-3.5, GPT-4).

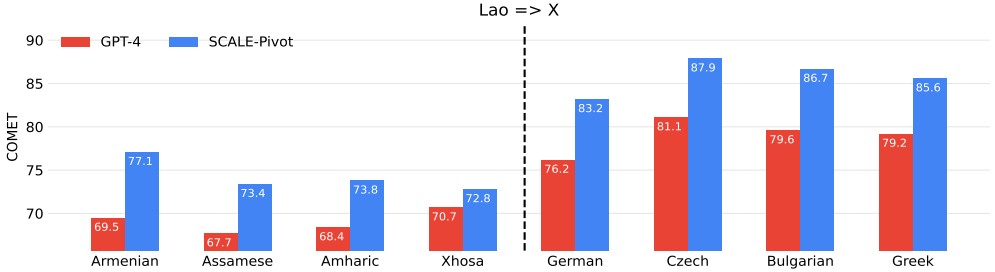

Figure 3: Translation results from Lao to both low- and high-resource languages, where GPT-4 uses few-shot prompting and SCALE-pivot uses English as the pivot language.

(e.g., SCALE-refine over GPT-4), the lexical metric spBLEU aligns with COMET and BLEURT. However, when comparing SCALE-refine with NLLB, although COMET-22, COMETkiwi, and BLEURT exhibit consistent patterns, spBLEU displays degradation with the GPT-based system in 4 out of 8 directions. Similar findings are also reported in Vilar et al. (2023); Hendy et al. (2023).

## 4.2 SCALE PIVOTING

In this section, we demonstrate the performance of SCALE-pivot, in which the variable $\mathbb{Z}$ is not directly pertinent to the current translation directions but functions as a pivot. Specifically, we examine the performance of few-shot GPT-4 and SCALE-pivot on Lao→ $\mathbb{Y}$ translations, where $\mathbb{Y}$ represents a language set encompassing both low-resource and high-resource languages. For the low-resource languages, we include Assamese (asm_Beng), Armenian (hye_Armn), Amharic (amh_Ethi), Xhosa (xho_Latn), and we have German (deu_Latn), Czech (ces_Latn), Bulgarian (bul_Cyrl) and Greek (ell_Grek) for the high-resource setting.

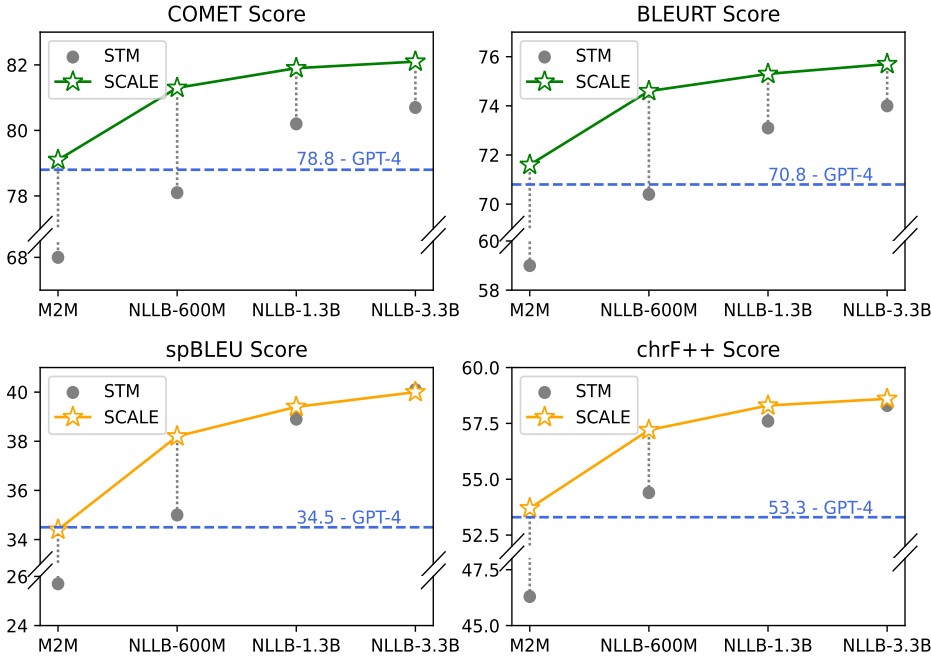

Figure 4: Translation results from Xhosa→English with evolving STMs in the SCALE framework.

The results of our analysis are presented in Figure 3. Upon examining the GPT-4 results in isolation, it is evident that the language bias inherent in the LLM has a considerable impact on translation performance. In particular, the few-shot GPT-4 model generally performs exceptionally well in high-resource settings; however, it tends to struggle in low-resource scenarios.

Moreover, our findings highlight that employing SCALE-pivot can effectively enhance the performance of GPT-4 across both low- and high-resource settings. Interestingly, the performance gains achieved through SCALE-pivot are more pronounced in high-resource settings, with an average improvement of 6.8 COMET-22 score compared to 5.2 for low-resource settings. This outcome suggests that incorporating SCALE-pivot can further boost the already strong performance of GPT-4 in high-resource situations, while also providing a notable improvement in low-resource contexts.

### 4.3 SCALE UPDATING

In this section, we explore the potential enhancement of our framework by keeping the LLM fixed and solely updating the STM. Specifically, we use M2M100-12B and NLLB model suite ranging from 600M to 3.3B as our evolving STM. We conduct experiments on the Xhosa → English direction and adopt the prompt format of SCALE-refine. The experimental results are displayed in Figure 4, leading to the following observations:

(1) The overall framework can be consistently improved with a fixed LLM and a continuously evolving STM; (2) SCALE, when equipped with a small model containing only 600M parameters, can outperform GPT-4 with an absolute 2.5 COMET-22 score and a 3.8 BLEURT score; (3) Equipped with an STM (M2M100) of relatively lower performance than original few-shot GPT-4 , SCALE demonstrates strong robustness by not merely copying and pasting the less satisfactory reference answer provided by M2M100, which we detailedly investigated in §5.3.

Interestingly, we also observe that the growth patterns exhibited by lexical metrics and neural semantic metrics differ. For M2M100 and NLLB-600M as STM, both metrics experience substantial improvement, while for NLLB-1.3B and 3.3B as STM, SCALE maintains the same lexical accuracy while continually enhancing translation performance as measured by neural semantic metrics.

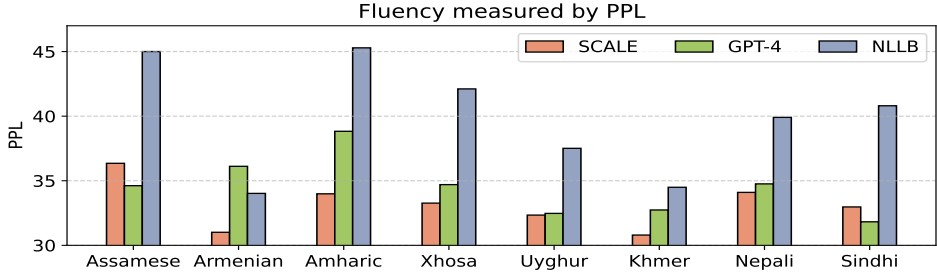

Figure 5: Perplexity score from $\mathbb{X} \rightarrow$English translation.

## 5 FURTHER ANALYSIS

### 5.1 TRANSLATION CHARACTERISTICS

To gain a deeper understanding of the translation characteristics of different systems (few-shot LLMs, STMs, and SCALE) beyond overall translation quality, we employ the following measurements, as suggested by Hendy et al. (2023):

1. **Translation Fluency:** Since LLMs are optimized by predicting the next token, their translations tend to display a language modeling bias that favors fluency over adequacy. To investigate this, we utilize an independently trained open-source language model (GPT2-XL (Radford et al., 2019)) to measure the perplexity score of the translation output.

2. **Translation Non-Monotonicity:** This metric evaluates the extent to which a translation adheres to the source sentence's structure, calculating the deviation from the diagonal in the word-to-word alignment. Translations that are more paraphrastic or less literal tend to deviate from closely tracking the source word order across language pairs (Hendy et al., 2023). We apply the non-monotonicity metric proposed by Schioppa et al. (2021).

3. **Unaligned Source Words:** Another measure of literalness is the count of unaligned source words (Hendy et al., 2023; Raunak et al., 2023a). When accounting for quality, less literal translations are likely to include more words that do not align with those in the source sentence.

We present the **Translation Fluency** results of $\mathbb{X} \rightarrow$ English translation in Figure 5, where $\mathbb{X}$ remains the same as used in Section 4.1. It is evident that regardless of the translation quality delivered by the LLM, whether superior (SCALE) or inferior (GPT-4) compared to the STM (NLLB), the LLM translation generally demonstrates higher fluency than the STM. Additionally, in 6 out of the 8 languages examined, SCALE produces lower perplexity scores than the original GPT-4 output. This suggests that the STM-generated variable $\mathbb{Z}$ can effectively aid the GPT-4 model in further decreasing its generation uncertainty.

For **Non-Monotonicity** and **Unaligned Source Words**, we choose Xhosa$\rightarrow$English translation with different STMs, and the results are shown in Figure 6. We also include PPL score for completeness. We find that both the USW and NM scores for STM are higher than those of GPT-4. This indicates that even though STM provides higher translation quality, it results in less literal translations. However, for SCALE, it effectively reduces GPT-4's NM score while maintaining a moderate USW score. This suggests that during the SCALE refinement process, the model primarily adheres to the original LLM output structure while taking cues from STM's word selection. We also show several concrete cases in Appendix A.3.

### 5.2 MULTIPATH SAMPLING

In this section, We list the results of multi-path sampling strategy in Table 2. We test with Xhosa$\rightarrow$English with one-shot SCALE-refine. The results show that without increasing the shot number in the few-shot learning, using STM to generate more generation paths could consistently

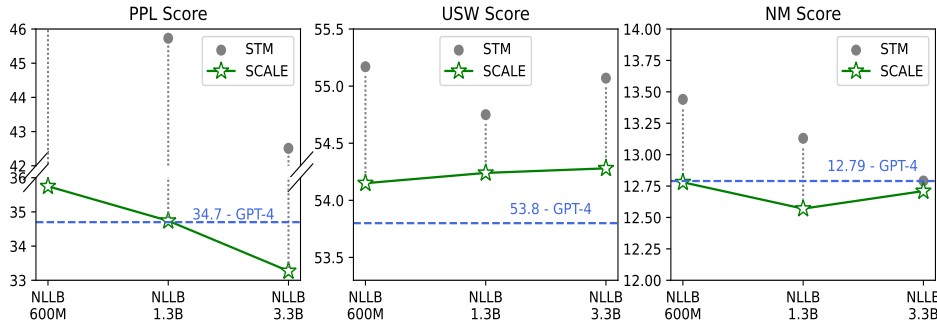

Figure 6: Perplexity, Unaligned Source Words percentage and Non-Monotonicity score from Xhosa→English translation.

| # Path | COMET-22 | BLEURT | spBLEU |
|--------|----------|--------|--------|
| 1 | 80.4 | 73.2 | 35.6 |
| 2 | 81.2 | 74.3 | 37.1 |
| 3 | 81.4 | 74.7 | 38.0 |
| 4 | 81.5 | 74.8 | 38.3 |
| 5 | 81.4 | 74.9 | 38.4 |

Table 2: Translation results from Xhosa→English with multi-path sampling. All the experiments are conducted by one-shot SCALE-refine and only differ in the number of sampled paths from STM.

improve the overall performance, which could be useful in the extremely low-resource setting where demonstration samples are hard to acquire.

### 5.3 ABLATION

In this section, we conduct an ablation study for each key design in our framework. We examine the following variants: (1) without confidence: This model follows the same setting as the SCALE-refine in §4.1, except that we do not pass the confidence score of each token as input. (2) zero-shot: This variant removes all in-context-learning examples, keeping only the translation instruction and the reference answer from STM. (3) one-shot: This model utilizes only one-shot, in contrast to the ten-shot results presented in §4.1. (4) zero-shot-M2M: This model also implements zero-shot, but the STM used is M2M100, a less performant model than the original few-shot GPT-4. This is employed to assess the robustness of our framework.

The outcomes of our ablation study are showcased in Table 3. It is evident that each component in our framework perform effectively, with the in-context-learning setting providing the most performance gain. This indicates that simply offering a reference answer to the LLM without in-context samples does not adequately guide the model in utilizing those references effectively. Furthermore, the number of ICL examples is also an essential factor in the process.

Regarding the SCALE zero-shot-M2M variant, its performance is significantly inferior to that of the few-shot LLM due to the poor quality of the M2M100 output. From this observation, we can conclude that the robustness of SCALE, as illustrated in Figure 4, primarily stems from the power of in-context learning. This learning approach informs the LLM about which elements to trust and which to disregard, ultimately improving the overall translation performance and robustness.

### 5.4 GENERATION LATENCY

In this section, we conduct a detailed evaluation of the overhead introduced by SCALE in comparison to few-shot LLM. The additional latency arises from two factors: first, the time required to generate the variable $\mathbb{Z}$ for the current source sentence $x$ using STM, and second, the increased latency caused by the LLM due to the extended context. We utilize one of the largest open-source LLMs (BLOOM-176B) for this analysis. As shown in Table 4, we observe that the incurred latency

|  | COMET-22 | COMETKiwi | BLEURT |
|---|---|---|---|
| M2M100 | 68.0 | 62.1 | 59.0 |
| NLLB | 80.7 | 65.8 | 74.0 |
| GPT-4 | 78.8 | 67.1 | 70.8 |
| SCALE | 82.1 | 67.3 | 75.7 |
| w/o confidence | 81.6 | 67.6 | 74.9 |
| zero-shot | 81.4 | 66.4 | 74.8 |
| one-shot | 81.7 | 66.7 | 75.3 |
| zero-shot-M2M | 76.4 | 66.8 | 68.2 |

Table 3: Ablation study for SCALE with Xhosa→English translation.

|  | few-shot LLM | | SCALE | | | |
|---|---|---|---|---|---|---|
|  | avg. #length | total | avg. #length | STM | LLM | total |
| 0-shot | 101.37 | 7.19 | 161.13 | 1.87 | 7.43 | 9.3 |
| 1-shot | 198.00 | 7.46 | 516.92 | 1.87 | 8.33 | 10.2 |
| 10-shot | 951.91 | 9.52 | 2489.72 | 1.87 | 14.17 | 16.04 |

Table 4: Generation latency results of LLM (BLOOM-175B) and SCALE (BLOOM-175B + NLLB-3.3B) measured in seconds (s).

can be primarily attributed to the extended context window due to the quadratic time complexity of the transformer. Exploring methods to accelerate this process based on STM-generated output using speculative decoding techniques remains future work (Xia et al., 2022; Yang et al., 2023).

## 6 RELATED WORK

The use of LLM for translation tasks has garnered significant interest in recent times. Brown et al. (2020) initially demonstrated the efficacy of prompting an LLM with a few examples to achieve note-worthy results, particularly in high-resource languages (Vilar et al., 2023; Lin et al., 2022). Following the release of ChatGPT, several studies have examined its overall translation performance(Jiao et al., 2023; Hendy et al., 2023), along with works focusing on the issue of hallucination (Guerreiro et al., 2023) , literalness (Raunak et al., 2023a), multilinguality (Zhu et al., 2023) and incidental bilingualism problem (Briakou et al., 2023). A comprehensive analysis conducted by Garcia et al. (2023) revealed the unreasonable effectiveness of few-shot LLMs. Furthermore, a diverse range of research has attempted to enhance LLM-based translation systems through cultural awareness (Yao et al., 2023), refinement (Chen et al., 2023; Cheng et al., 2023b), retrieval-augmentation (Cheng et al., 2023b), post-editing (Raunak et al., 2023b), and comparison (Zeng et al., 2023).

Our work also shares similarities with a series of studies that aim to build collaboration between LLMs and other systems. Luo et al. (2023) propose equipping LLMs with a knowledge-guiding module to access relevant information without altering the LLMs' parameters. Hendy et al. (2023) propose to use Microsoft Translator system as the primary translation system, and then use GPT as a fallback system when the quality of MS-Translator is unsatisfactory measured by reference-free metrics. Xu et al. (2023) introduce SuperICL and achieve significant improvements in various language understanding tasks. Ge et al. (2023) employ a trainable LoRA-based encoder as an additional model for LLM context compression.

## 7 CONCLUSION

In this paper, we present a novel framework SCALE, which effectively combines the strengths of Large Language Models (LLMs) and compact Specialized Translation Models (STMs) through in-context learning. By providing triplet in-context demonstrations, our framework successfully unlocks the refinement and pivoting capabilities of LLMs. SCALE demonstrates its superiority in many scenarios including low-resource setting, multilingual translation and model continual learning setting. Our results offer crucial understanding and a robust basis for subsequent research investigating the possible synergistic effects between LLMs and more specialized translation models.

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

# A APPENDIX

## A.1 PROMPT EXAMPLE

In Table 5, we list the prompt we use for few-shot LLM and in Table 6, for our SCALE framework. We use Chat Markup Language version from Azure to format our prompt[1].

| | |
|---|---|
| Instruction | < \|im_start\| >system
Assistant is an intelligent chatbot designed
to help users translate from ${source_language} to ${target_language}
< \|im_end\| > |
| Examples | < \|im_start\| >user
Source: ${source_1}
Target: ${target_1}
...
Source: ${source_n}
Target: ${target_n} |
| Input | Source: ${source}
< \|im_end\| >
< \|im_start\| >assistant
Target: |

Table 5: Prompt of Chat Markup Language format for few-shot LLM.

| | |
|---|---|
| Instruction | < \|im_start\| >system
Assistant is an intelligent chatbot designed
to help users translate from ${source_language} to ${target_language}

Context:
· Assistant would would be given a potentially useful reference  answer
from a fine-tuned model
· The number in brackets denotes the confidence score of a fine-tuned model
to generate the token.
< \|im_end\| > |
| Examples | < \|im_start\| >user
Source: ${source_1}
Potentially useful reference answer 1: ${reference_1}
Potentially useful reference answer 2: ${reference_2}
Target: ${target_1}
...
Source: ${source_n}
Potentially useful reference answer 1: ${reference_1}
Potentially useful reference answer 2: ${reference_2}
Target: ${target_n} |
| Input | Source: ${source}
Potentially useful reference answer 1: ${reference_1}
Potentially useful reference answer 2: ${reference_2}
< \|im_end\| >
< \|im_start\| >assistant
Target: |

Table 6: Prompt of Chat Markup Language format for SCALE.

---

[1]https://learn.microsoft.com/en-us/azure/ai-services/openai/how-to/chatgpt?pivots=programming-language-chat-ml

## A.2 Data Statistics

We list the detailed data information for SCALE-refine and SCALE-Pivot experiments in Table A.2. The number of dev set is 997 and 1012 for devtest set in flores-200 (NLLB Team et al., 2022).

| code | language | # dev length | # devtest length | script | family | resource |
|------|----------|--------------|------------------|--------|--------|----------|
| asm_Beng | Assamese | 40.55 | 41.67 | Bengali | Indo-European | low |
| hye_Armn | Armenian | 43.91 | 45.31 | Armenian | Indo-European | low |
| amh_Ethi | Amharic | 38.87 | 39.64 | Ge'ez | Afro-Asiatic | low |
| xho_Latn | Xhosa | 35.31 | 36.37 | Latin | Atlantic-Congo | low |
| uig_Arab | Uyghur | 40.77 | 42.41 | Arabic | Turkic | low |
| khm_Khmr | Khmer | 52.77 | 53.79 | Khmer | Austroasiatic | low |
| npi_Deva | Nepali | 34.36 | 35.48 | Devanagari | Indo-European | low |
| eng_Latn | English | 28.99 | 30.28 | Latin | Indo-European | high |
| deu_Latn | German | 37.57 | 39.16 | Latin | Indo-European | high |
| ces_Latn | Czech | 36.63 | 38.10 | Latin | Indo-European | high |
| bul_Cyrl | Bulgarian | 37.99 | 39.45 | Cyrillic | Indo-European | high |
| rus_Cyrl | Russian | 39.42 | 40.21 | Cyrillic | Indo-European | high |

Table 7: Data statistics for all the tested languages in the paper.

## A.3 Translation Cases

In this section, we list several translation cases from different languages.

| | |
|---|---|
| **SOURCE** | बाइसन, एल्क, मूस, भालु र लगभग सबै ठूला जनावरहरूले जस्ता नरम देखिए पनि आक्रमण गर्न सक्छन्। |
| **TARGET** | No matter how docile they may look, bison, elk, moose, bears, and nearly all large animals can attack. |
| **MS Translator** | Bison, elk, moose, bears, and almost all large animals can attack even if they look soft. |
| **NLLB** | The Bible says: "The one who is walking with wise persons will become wise, but the one who is having dealings with the stupid ones will fare badly". |
| **GPT-4** | Bison, elk, moose, bears, and nearly all large animals, despite appearing gentle, can be aggressive. |
| **SCALE** | Bison, elk, moose, bears and nearly all large animals can attack even though they appear docile. |

Figure 7: Translation case from Nepali→English.

| | |
|---|---|
| **SOURCE** | ভৰি থোৱা ৰিকাবে চলাওঁতাজনৰ ভৰি ৰখাত সহায় কৰে যিটো ঘোঁৰাৰ গা-দীৰ দুয়োফালে তললৈ ওলমি থাকে। |
| **TARGET** | Stirrups are supports for the rider's feet that hang down on either side of the saddle. |
| **MS Translator** | The legged rickshaw helps to keep the driver's leg which hangs down on either side of the horse's mattress. |
| **NLLB** | The foot rest helps to keep the rider's feet which are sloping downwards on both sides of the horse's saddle. |
| **GPT-4** | A heavily loaded Rickshaw helps balance the load by tilting to both sides when going over bumps. |
| **SCALE** | The stirrup helps to support the rider's feet, which are sloping downwards on both sides of the horse's saddle. |

Figure 8: Translation case from Assamese→English.

| | |
|---|---|
| **SOURCE** | የዳይኖሰር ካባዎች የዳበረ ራኺስ የሚባል ዘንግ ስለሌለው፣ ነገር ግን ሌሎች የላባ ባህርያት — ባርብስ እና ባርቡልስ — ስላሉ ተመራማሪዎች ራኺስ ከእነዚህ ሌሎች ባህርያት የቆየ ዝግመተ ለውጥ ውጤት እንደሆነ ዮላሉ። |
| **TARGET** | Because the dinosaur feathers do not have a well-developed shaft, called a rachis, but do have other features of feathers — barbs and barbules — the researchers inferred the rachis was likely a later evolutionary development that these other features. |
| **MS Translator** | Dinosaur feathers developed because it doesn't have a rod called rachis, but has other feather traits — barbs and barbules — that researchers say is the result of older evolution of rachis from these other traits. |
| **NLLB** | dinosaur feathers did not develop a shaft called the rachis, but other feather features, such as barbs and barbels, suggest that the rachis was the result of an earlier evolution of these other features. |
| **GPT-4** | As there is no known population of the extinct Laysan Rail on Laysan Island, researchers suggest that the presence of rails on the other islands—Barbados and Barbuda—indicates a prolonged period of isolation and change. |
| **SCALE** | Dinosaur feathers did not develop a shaft called the rachis, however, other feather features such as barbs and barbules suggest that the rachis was the result of an earlier evolution of these other features. |

Figure 9: Translation case from Amharic→English.

| SOURCE | बाइसन, एल्क, मूस, भालु र लगभग सबै ठूला जनावरहरूले जस्ता नरम देखिए पनि आक्रमण गर्न सक्छन्। |
|---|---|
| TARGET | Auch das Tragen eines Rings ist hilfreich (nur keinen, der zu teuer aussieht |
| GPT-4 | Es gibt eine Chance, dass es genauso verschwindet, wie es aussieht, als ob es einfach verschwindet. |
| SCALE | Es ist auch nützlich, einen Ring zu tragen, nur scheint der Ring zu teuer zu sein. |

Figure 10: Translation case from Lao→German.

