# OpenReview forum: "SCALE: Synergized Collaboration of Asymmetric Language Translation Engines"
_ICLR.cc/2024/Conference — Submitted to ICLR 2024_

### Official Review · Reviewer_xCiG · 2023-10-29

**Soundness:** 3 good
**Presentation:** 4 excellent
**Contribution:** 2 fair
**Rating:** 5
**Confidence:** 5

**Summary:**

This paper introduces the SCALE framework, which integrates specialized Translation Models (STM) with Large Language Models (LLMs) such as GPT-4, utilizing triplet examples for few-shot learning. Each triplet incorporates an additional element: a translation generated by the STM, supplementing the standard source and target sentences. The study demonstrates significant enhancements in translating from various low-resource languages into English (X→En). Furthermore, the paper conducts ablation studies to show the individual contributions of each SCALE component to the overall performance improvement.

**Strengths:**

- The paper is easy to understand and well written.
- The idea is simple but effective. The SCALE framework shows substantial improvement for low-resource languages when translating into English.
- The paper did a good ablation study to show the essential role of each component in the SCALE framework.

**Weaknesses:**

Major weaknesses:
- I like the idea; however, the paper focuses on translation into English and does not demonstrate the efficacy of translating from English to these low-resource languages. I suspect that SCALE has limitations in translating from English, given that refinement is core to SCALE. Although GPT-4 excels in English, it falls short in low-resource languages, suggesting that SCALE may not be effective for En→X.
- SCALE seems confined to low-resource languages, as LLM already performs excellently in translating between high-resource languages, such as German and French. The paper should make it clearer for readers how SCALE fares in the context of translating high-resource languages.
- In presenting "pivoting" (Sec. 4.2) and "updating" (Sec. 4.3), the paper restricts its experiments to LAO→En and Xhosa→En, respectively, which doesn't sufficiently substantiate this aspect. Conducting more comprehensive experiments would provide stronger support for the concept.

Minor weaknesses:
- The paper's assertion that SCALE "mitigates both language bias and parallel data bias" seems peculiar to me. It feels intuitive rather than conclusively factual. Defining "language bias" and "parallel data bias," followed by a quantitative analysis, might be necessary for clarity.
- The choice of a less advanced model like GPT-2 for evaluating GPT-4's fluency strikes me as odd. While employing GPT-2 might yield reasonable comparative results for less sophisticated generation models, but I do not think it is proper here.

**Questions:**

I would be appreciate if the authors can address my concerns above!

---

> ### Author Response · Authors · 2023-11-19
>
> Dear Reviewer,
>
> We sincerely appreciate your time and insightful comments on our paper. Your commendation of our idea is encouraging, and we are eager to address the concerns you raised. Further, we kindly request that you reconsider your score should our responses help alleviate some of your concerns.
>
> Concern #1: The efficacy of the En→X translation setting
>
> Response: We appreciate your insightful comment and agree that evaluating the efficacy of En→X translation is crucial. To address this, we have expanded our experiments to include translations from English to six low-resource languages, using the same experimental setup as that of SCALE-refine. The results, gauged by COMET-22 and COMETKiwi, are presented below, and all corresponding outputs have been appended to the supplementary zip files for your reference. These additional experiments lead us to several observations:
>
> 1. SCALE exemplifies its proficiency in translating from English to a diverse range of languages, spanning various scripts and linguistic families.
> 2. While SCALE shows improvement over few-shot GPT-4 and NLLB, the magnitude of this improvement is less pronounced compared to the X→En setting. This indicates that the present limitation for LLM translators mainly lies in the generation of the target language, rather than the comprehension of the source language, as also suggested in [1].
> 3. Due to the limited transparency of the GPT-4 model, it is challenging to attribute these inconsistencies to specific factors like tokenization, data coverage, or the proportion of languages in the training data. Regardless, we are striving to identify a potential indicator, specifically the number of native speakers, to illustrate the extent of a language's resources. As illustrated in the accompanying table, the performance of few-shot GPT-4 diminishes with the number of native speakers, while our framework, SCALE, consistently and effectively mitigates this language bias, outperforming few-shot GPT-4 and the supervised NLLB model in 6/6 directions.
>
> Recognizing the significance of these findings, we would include this discussion in the final version of our paper.
>
> |                   |   Javanese    |     Tamil     |    Urdu     |    Amharic    |     Khmer     |    Nepali     |
> | --------------- | :-----------: | :-----------: | :---------: | :-----------: | :-----------: | :-----------: |
> | # Native Speakers |     98 M      |    84.12 M    |   71.29 M   |     25 M      |     18.4M     |     16 M      |
> |  few-shot GPT-4   |   83.9/75.8   |   83.5/80.8   |  80.0/80.4  |   77.1/73.4   |   74.3/74.7   |   80.1/86.4   |
> |     NLLB-3.3B     |   86.4/76.4   |   86.5/82.9   |  80.7/80.4  |   84.4/80.7   |   76.3/77.8   |   77.0/82.0   |
> |       SCALE       | **86.6/77.5** | **87.8/84.7** | **82/81.7** | **84.7/81.7** | **79.6/80.3** | **82.6/87.8** |
>
> ---
> Concern #2: Clear motivation for low-resource setting.
>
> Response: We acknowledge your concern about the perceived focus of SCALE on low-resource languages and apologize if our manuscript did not communicate this aspect clearly. Your suggestion to clarify the motivation of our experimental design is well taken.
>
> While it is indeed true that current LLMs demonstrate exceptional performance in high-resource settings, our curiosity was piqued regarding the performance of SCALE in high-resource languages. We carried out experiments in three languages: German, Czech, and Chinese, under the same settings as SCALE-refine. We utilized COMET-22 and COMETKiwi to measure the translations. As shown below, the results reveal that SCALE offers consistent improvements even for high-resource languages, albeit less significantly than for low-resource languages, and it exhibits robust performance even when paired with a less performant STM (especially in en-zh direction). This is remarkable, given the already strong performances of models such as GPT-4 and NLLB. All the generated output has been updated to the supplementary zip files for your reference.
>
> We assure you that these insights will be integrated into the revised version of our manuscript to provide a more comprehensive understanding.
>
> |                |     en-de     |     de-en     |     en-cs     |     cs-en     |     en-zh     |     zh-en     |
> | ------------ | :-----------: | :-----------: | :-----------: | :-----------: | :-----------: | :-----------: |
> | few-shot GPT-4 |   88.5/84.4   | **90.0**/84.6 |   92.0/86.8   |   88.4/85.0   | 88.8/**84.7** |   87.3/84.3   |
> |   NLLB-3.3B    |   86.9/82.9   |   89.1/84.2   |   90.1/84.8   |   88.4/85.5   |   78.0/70.9   |   86.1/83.7   |
> |     SCALE      | **89.0/85.1** | 89.9/**84.7** | **92.4/87.1** | **89.2/86.0** | **89.1/84.7** | **87.8/84.9** |

---

> > ### Author Response · Authors · 2023-11-19
> > **Response to Reviewer xCiG (continued)**
> >
> > Concern #3: Limited range of experiments for "pivoting" (Sec. 4.2) and "updating" (Sec. 4.3).
> >
> > Response:  We appreciate your observation regarding the limited range of experiments for the "pivoting" and "updating" sections. However, it is important to clarify that SCALE in the pivoting section was not limited to Lao→En translations. We expanded the translation from Lao to 8 other languages, encapsulating both low- and high-resource settings. Since the language selection in both the pivoting and updating sections was not specifically curated, we believe that this showcases the versatility of our method.
> >
> > In accordance with your suggestion for a more exhaustive set of experiments, we have conducted additional experiments in both pivoting and updating settings. These new experiments include a broader range of languages, offering robust support for our concept. For the pivoting setting, we extended our experiments from Xhosa to eight other languages, covering both low- and high-resource languages, employing both few-shot GPT-4 and SCALE-pivot. We utilized COMET-22 to measure the performance. The results of these tests are presented below and we observe a similar pattern as exhibited in "pivoting" (Sec. 4.2). All the generated output has been updated to the supplementary zip files for your reference.
> >
> > |                | Armenian | Assamese | Amharic  |   Lao    |  German  |  Czech   | Bulgarian |  Greek   |
> > | -------------- | :------: | :------: | :------: | :------: | :------: | :------: | :-------: | :------: |
> > | # Resource     |   Low    |   Low    |   Low    |   Low    |   High   |   High   |   High    |   High   |
> > | few-shot GPT-4 |   68.3   |   65.6   |   69.3   |   58.5   |   74.4   |   80.8   |   79.1    |   78.6   |
> > | SCALE-pivot    | **71.7** | **67.4** | **71.4** | **59.6** | **77.9** | **83.7** | **82.7**  | **81.4** |
> >
> > For updating setting, we have broadened our scope to include three additional languages: Lao, Armenian, and Amharic. To assess the translations, we utilized the COMET-22 and COMETKiwi evaluation metrics. The results, which exhibit a pattern similar to that described in the "updating" section (Sec. 4.3), are presented below. For your reference, we have also updated the supplementary zip files with all generated outputs.
> >
> > |                |           |     Xhosa     |      Lao      |   Armenian    |    Amharic    |
> > | :------------- | :-------: | :-----------: | :-----------: | :-----------: | :-----------: |
> > | STM            |    M2M    |   68.0/59.0   |   67.8/57.5   |   75.9/58.9   |   72.3/54.8   |
> > |                | NLLB-600M |   78.1/70.4   |   84.6/70.3   |   86.3/73.4   |   84.7/69.2   |
> > |                | NLLB-1.3B |   80.7/74.0   |   86.9/73.8   |   87.7/75.6   |   86.2/71.9   |
> > |                | NLLB-3.3B |   80.7/74.0   |   86.9/73.8   |   88.3/77.0   |   86.9/73.6   |
> > | few-shot GPT-4 |           |   78.8/70.8   |   80.0/63.7   |   86.2/73.1   |   83.2/67.3   |
> > | SCALE w/       |    M2M    |   79.1/71.6   |   82.5/67.3   |   86.7/74.1   |   84.6/69.7   |
> > |                | NLLB-600M |   81.3/74.6   |   86.3/72.9   |   88.1/76.3   |   87.3/74.2   |
> > |                | NLLB-1.3B |   81.9/75.3   |   86.6/73.5   |   88.5/77.0   |   87.8/75.1   |
> > |                | NLLB-3.3B | **82.1/75.7** | **87.2/74.4** | **88.8/77.8** | **88.0/75.7** |
> >
> > ---
> > Concern #4: The paper's claim that SCALE "mitigates both language bias and parallel data bias”
> >
> > Response:  We appreciate your feedback regarding the need for explicit definitions and comprehensive analysis of the terms "language bias" and "parallel data bias". We regret any confusion that may have been caused by their absence in our initial submission.
> >
> > When referring to "language bias", we are addressing the overrepresentation of English in the pre-training corpus of LLMs. This often leads to superior performance in English while underperforming in other languages, particularly those with limited resources. We have conducted experiments involving refinement and pivoting, which have shown that SCALE provides more equitable performance across these underrepresented languages. We interpret this as an effective mitigation of the inherent language bias present in models such as GPT-4.
> >
> > On the other hand, "parallel data bias" pertains to the issues faced by supervised models when dealing with substandard or skewed samples from coarsely-mined parallel data [2]. Regarding SCALE, as the final translation is not generated by a STM, we posit that SCALE has the potential to alleviate this bias to a considerable extent. We present empirical evidence in our robustness analysis, demonstrating that SCALE maintains its effectiveness even when less satisfactory translations are provided by a STM.
> >
> > We hope that these clarifications address your concerns, and we thank you for your constructive feedback.

---

> ### Author Response · Authors · 2023-11-19
> **Response to Reviewer xCiG (continued)**
>
> Concern #5: The selection of GPT-2 for experiments.
>
> Response: We understand your concerns about our choice of using GPT-2 in our experiments. The use of GPT-2 in this context is solely as an evaluator to measure the fluency of translations, a role it has successfully fulfilled in recent studies such as [2]. Despite being a model from four years ago, GPT-2 still serves as an effective evaluator for assigning perplexity scores to English sentences.
>
> To address your concern and ensure the reliability of our results, we also conducted the same experiments using Llama-2-7B and Llama-2-13B. The results of these tests are shown below. While Llama2 gives different absolute values of perplexity of STM, GPT-4, and SCALE, the relative order remains identical to that obtained with GPT-2, thereby reinforcing our confidence in the original findings.
>
> This is the original results produced by GPT-2-XL:
>
> |       | Assamese  | Armenian  |  Amharic  |   Xhosa   |  Uyghur   |   Khmer   |  Nepali   |  Sindhi   |
> | ----- | :-------: | :-------: | :-------: | :-------: | :-------: | :-------: | :-------: | :-------: |
> | NLLB  |   45.00   |   34.02   |   45.29   |   42.11   |   37.51   |   34.49   |   39.90   |   40.80   |
> | GPT-4 | **34.61** |   36.11   |   38.83   |   34.70   |   32.47   |   32.73   |   34.75   | **31.82** |
> | SCALE |   36.35   | **31.01** | **33.98** | **33.26** | **32.34** | **30.79** | **34.09** |   32.97   |
>
> This is the results produced by Llama-2-7B:
>
> |       | Assamese  | Armenian  |  Amharic  |   Xhosa   | Uyghur |   Khmer   |  Nepali   |  Sindhi   |
> | ----- | :-------: | :-------: | :-------: | :-------: | :----: | :-------: | :-------: | :-------: |
> | NLLB  |   27.08   |   21.39   |   26.88   |   25.72   | 24.53  |   22.70   |   23.74   |   25.32   |
> | GPT-4 |   21.77   |   21.18   |   23.64   |   21.26   | 20.50  |   20.63   |   21.33   | **19.90** |
> | SCALE | **21.68** | **18.94** | **21.20** | **20.82** | **20.29**  | **19.63** | **20.24** |   20.37   |
>
> This is the results produced by Llama-2-13B:
>
> |       | Assamese  | Armenian  |  Amharic  |   Xhosa   |  Uyghur   |   Khmer   |  Nepali   |  Sindhi   |
> | ----- | :-------: | :-------: | :-------: | :-------: | :-------: | :-------: | :-------: | :-------: |
> | NLLB  |   27.41   |   21.37   |   26.73   |   26.01   |   24.74   |   22.90   |   23.68   |   25.42   |
> | GPT-4 | **22.01** |   21.53   |   23.75   |   21.01   |   20.58   |   20.68   |   21.23   | **20.07** |
> | SCALE |   22.11   | **18.84** | **21.21** | **20.63** | **20.49** | **19.80** | **20.12** |   20.41   |
>
> [1] Raunak, Vikas, et al. "Leveraging GPT-4 for Automatic Translation Post-Editing." *arXiv preprint arXiv:2305.14878* (2023).
>
> [2] Hendy, Amr, et al. "How good are gpt models at machine translation? a comprehensive evaluation." *arXiv preprint arXiv:2302.09210* (2023).

---

> ### Comment · Reviewer_xCiG · 2023-11-20
> **Thank authors for rebuttal**
>
> I appreciate the authors for providing comprehensive results. This indeed makes me better understand how SCALE works on en->xx and high-resource languages. It surprises me that SCALE also works well on en->xx (though less pronounced). Although this method is not very helpful for high-resource languages, I believe this paper should be interesting to researchers interested in low-resource translation. Thus, I have raised my score. The authors seem to conduct extra experiments for four new low-resource languages (only Amharic and Nepali overlap with the paper), not the original eight low-resource languages, i.e., Assamese (asm Beng), Armenian (hye Armn), Amharic (amh Ethi), Xhosa (xho Latn), Uyghur (uig Arab), Khmer (khm Khmr), Nepali (npi Deva), and Sindhi (snd Arab). Why is this?

---

> > ### Author Response · Authors · 2023-11-21
> >
> > Dear Reviewer,
> >
> > Firstly, we would like to extend our gratitude for the time and effort you've invested in reviewing our rebuttal. Your recognition and improved scoring are greatly appreciated. However, we feel the need to clarify certain aspects further:
> >
> > 1. Our work should not be confined to the perception of mere enhancement for low-resource languages. Fundamentally, our research probes an essential question in machine translation within the era of Large Language Models: Should we persist in training specialized translation models, or should we fully embrace LLM-based translations?  LLMs have demonstrated a general proficiency that extends beyond machine translation, but they face challenges with language bias and the non-trivial nature of updating the model in terms of computational resources and the balance between generality and speciality. On the other hand, STMs only excel in a pre-defined task and may be impeded by the parallel data bias inherent in supervised training. Our study pioneers a novel approach by proposing a viable solution that merges these two systems within one unified framework. Our paper elucidates not only the efficacy of SCALE-refine in a low-resource setting, but it also showcases the great potential of SCALE when integrated with evolving STMs of varying quality with SCALE-update and demonstrates its general translation ability between any language pairs with SCALE-pivot. Importantly, as the LLM in our framework does not require additional training, it does not compromise the general abilities of the LLM and stands to benefit from further improvements from either STMs or LLMs.
> >
> > 2. We regret the oversight in our initial rebuttal regarding the rationale behind our selection of the six demonstration languages. These languages were chosen based on our belief that the existing classification dividing all languages into two categories, low- and high-resource, is somewhat restrictive and coarse-grained. As illustrated in the Table, the last eight columns represent the languages used in our paper, while the first three are recent additions. Despite all being classified as low-resource languages according to [1], they are categorized differently according to \[2][3] and vary significantly in terms of the number of native speakers, a factor that potentially indicates the extent of available training data. Since the eight initial languages are extremely low-resource (each with fewer than 25M native speakers), we sought to explore how language bias could affect the performance of black-box GPT-4 and SCALE. To this end, we selected three additional languages with slightly higher resources, namely Javanese, Tamil, and Urdu.
> >
> > |                                            | Javanese* | Tamil*  |  Urdu*  | Amharic* | Sindhi | Khmer* | Nepali* | Assamese | Uyghur | Xhosa | Armenian |
> > | :----------------------------------------: | :-------: | :-----: | :-----: | :------: | ------ | ------ | :-----: | :------: | ------ | :---: | :------: |
> > | Category from Costa-jussà, Marta R., et al |    Low    |   Low   |   Low   |   Low    | Low    | Low    |   Low   |   Low    | Low    |  Low  |   Low    |
> > |    Category from Joshi, Pratik, et al.     |     1     |    3    |    3    |    2     | 1      | 1      |    1    |    1     | 1      |   2   |    1     |
> > |             # Native Speakers              |   98 M    | 84.12 M | 71.29 M |   25 M   | 24M    | 18.4M  |  16 M   |   15M    | 11M    | 10 M  |  5.3 M   |

---

> > > ### Author Response · Authors · 2023-11-21
> > >
> > > 3. To further address your concerns, we conducted experiments on the two lowest resource languages, Xhosa and Armenian, both of which have less than 10M native speakers. The results, shown below, reveal that SCALE still outperforms GPT-4 and NLLB, albeit limitedly with those slightly higher resource languages. Moreover, the language bias pattern persists for Xhosa, where GPT-4 only achieves a 72.3 COMET-22 score and 52.3 COMETKiwi score. However, in Armenian, both GPT-4 and SCALE perform better than languages with significantly more native speakers (e.g., Urdu). Due to the GPT-4 model's limited transparency and the approximation of this proxy indicator, pinpointing the reason behind this remains challenging, and we plan to explore this in our future work.
> > >
> > > |                                     |   Javanese    |     Tamil     |    Urdu     |    Amharic    |     Khmer     |    Nepali     |     Xhosa     |   Armenian    |
> > > | :---------------------------------- | :-----------: | :-----------: | :---------: | :-----------: | :-----------: | :-----------: | :-----------: | :-----------: |
> > > | Category from Joshi, Pratik, et al. |       1       |       3       |      3      |       2       |       1       |       1       |       2       |       1       |
> > > | # Native Speakers                   |     98 M      |    84.12 M    |   71.29 M   |     25 M      |     18.4M     |     16 M      |      9 M      |     5.3 M     |
> > > | few-shot GPT-4                      |   83.9/75.8   |   83.5/80.8   |  80.0/80.4  |   77.1/73.4   |   74.3/74.7   |   80.1/86.4   |   72.3/52.5   |   81.5/79.9   |
> > > | NLLB-3.3B                           |   86.4/76.4   |   86.5/82.9   |  80.7/80.4  |   84.4/80.7   |   76.3/77.8   |   77.0/82.0   | **75.2**/52.3 |   87.1/85.6   |
> > > | SCALE                               | **86.6/77.5** | **87.8/84.7** | **82/81.7** | **84.7/81.7** | **79.6/80.3** | **82.6/87.8** | 75.0/**53.4** | **87.6/85.8** |
> > >
> > >
> > >
> > > We sincerely thank you once more for your valuable review of our paper. We assure you that these insights will be incorporated into our manuscript's revised version to offer a more holistic understanding. We kindly request you to reconsider your score if our responses have addressed some of your concerns, and we stand ready to respond to any further queries you may have.
> > >
> > > [1] Costa-jussà, Marta R., et al. "No language left behind: Scaling human-centered machine translation." *arXiv preprint arXiv:2207.04672* (2022).
> > >
> > > [2] Joshi, Pratik, et al. "The state and fate of linguistic diversity and inclusion in the NLP world." ACL 2020.
> > >
> > > [3] Ranathunga, Surangika, et al. "Neural machine translation for low-resource languages: A survey." *ACM Computing Surveys* 55.11 (2023): 1-37.

---

> > > > ### Author Response · Authors · 2023-11-22
> > > >
> > > > Dear Reviewer xCiG,
> > > >
> > > > We would like to express our gratitude for the invaluable time and effort you have invested in reviewing our original paper and our subsequent rebuttal.
> > > >
> > > > As we near the conclusion of the author-reviewer discussion phase, scheduled to close within the next 24 hours, we greatly value the interactive dialogue this process fosters and the opportunity it provides to further enhance our work through your expertise.
> > > >
> > > > In light of our recent discussions, we seek your confirmation on whether our responses have sufficiently addressed your concerns. Please rest assured, we remain committed to engaging in further dialogues and we are prepared to address any additional queries you may have.
> > > >
> > > > Sincerely,
> > > >
> > > > Authors

---

> ### Comment · Reviewer_xCiG · 2023-11-22
>
> I thank the authors for their response. I am inclined to maintain my score for the following reasons:
>
> - The scope of the initial paper primarily focuses on xx->en translations in low-resource languages. While the authors have presented results for high-resource languages and en->xx translations, their effectiveness appears less pronounced.
> - I would be more inclined to rate the paper based on the initial submission. I acknowledge that some of results were shown in the rebuttal, but the manuscript could be substantially improved further with these extra experiments, e.g.,  en->xx translation results, results for high-resource languages, and a more comprehensive ablation study.
>
> Nonetheless, this work should be of interest to those engaged in LLM machine translation and research in low-resource languages.

---

> > ### Author Response · Authors · 2023-11-23
> >
> > Deeply appreciate your thoroughness and the thoughtful comments. Your feedback is instrumental in improving the quality of our work!

---

### Official Review · Reviewer_tFyE · 2023-10-30

**Soundness:** 3 good
**Presentation:** 3 good
**Contribution:** 3 good
**Rating:** 6
**Confidence:** 4

**Summary:**

This paper proposes a framework SCALE, to incorporate the translation ability of a specialized model into a LLM. The idea is to provide the results of a STM as extra input to the LLM, and let LLM generate new translations considering the input. This idea is quite simple and effective. It also works well in providing updating translations for refinement, pivoting and updating purposes.

**Strengths:**

The paper demonstrated a nice way of incorporating STM into LLMs.

The method is useful in three different senarios (refinement, pivoting and updating) in almost the same way.

According to the evaluation, the results are better than both the STM and LLMs.

**Weaknesses:**

My main concerns are in the analysis part:

For the analysis in 5.1, although the perplexity of a GPT2-XL decreases after SCALE refinement, there is no evidence that the results of larger LMs decreases as well. It is highly likely that the original results are more preferred by GPT4 than the refined results.

It is not straightforward to me why the STM results are with higher NM score than GPT4, but based on these results, SCALE achieves results with lower NM than GPT4.

It is not even clear for me whether the NM or USW should be higher or lower. According to the analysis, GPT4 produces results that are more literal (which is bad ), while STM have results more figurative. But it seems to me that GPT4 understand source languages better than STM, which is more likely to generate figurative translations.

Besides, since the LLMs learn how to use the latent variable Z by the examples, it might be extremely important to choose proper demonstrations. It will be useful to check the effects of different demonstrations.

**Questions:**

See above.

---

> ### Author Response · Authors · 2023-11-19
>
> Dear Reviewer,
>
> Thank you for your comprehensive feedback and for investing your time in reviewing our paper. We appreciate your insightful comments and understand your concerns. Allow us to provide some clarifications.
>
> Firstly, we would like to point out that the purpose of our Analysis section is to delve into the translation characteristics of the LLM, STM, and SCALE, rather than to present overall quality metrics such as COMET or BLEU. Regarding the perplexity score, we did not aim to decrease the perplexity of the GPT2-XL model. Instead, our intention was to utilize GPT2-XL as an evaluator to assess the fluency of the translations generated by these three translation systems. The conclusion drawn was that when STM-generated output is used as in-context demonstrations, GPT-4 has the potential to generate more fluent translations (similar pattern is also found when using Llama-2 as evaluator as shown in our response to Reviewer xCiG).
>
> We understand your confusion regarding the connection between fluency and quality. However, we abstained from establishing a direct correlation as LLM-based translations often exhibit higher fluency, irrespective of the quality of the translation. For a more detailed analysis, we refer to Section 5.2 in [1].
>
> Moving on to the NM and USW score, this metric is designed to measure the literalness of the translation output. Interestingly, the higher NM score observed for STM resulted in a lower NM score for SCALE. A conceivable explanation for this is that in the triplet in-context learning, the role of the STM-generated output is not to instruct the LLM to generate specific words, but rather to constrain the paraphrastic expression space, thereby helping the LLM to achieve a higher translation quality by comparing $Z$ and $Y$ in the triplet demonstrations. As mentioned in our paper, SCALE tends to maintain the original structure of the LLM output, whilst incorporating word selection cues from the STM.
>
> With respect to your question about the selection strategies for in-context learning samples, we did conduct some preliminary studies to evaluate their impact. Surprisingly, the performance of SCALE did not show a significant variation with the selection strategy. The three strategies we used were (1) first-k, (2) random, and (3) maximum-coverage, as suggested in [2]. No significant performance differences were observed among these three methods. We value your suggestion and will ensure this aspect is included in the final draft of our paper.
>
> We hope that our clarifications have satisfactorily addressed your concerns. We kindly request you to reconsider your score in light of these explanations. We welcome any further questions or concerns you may have and assure you that we are committed to improving our work based on your valuable feedback.
>
> Thank you once again for your time and effort.
>
> [1] Hendy, Amr, et al. "How good are gpt models at machine translation? a comprehensive evaluation." *arXiv preprint arXiv:2302.09210* (2023).
>
> [2] Sweta Agrawal, Chunting Zhou, Mike Lewis, Luke Zettlemoyer, and Marjan Ghazvininejad. 2023. [In-context Examples Selection for Machine Translation](https://aclanthology.org/2023.findings-acl.564). In *Findings of the Association for Computational Linguistics: ACL 2023*, pages 8857–8873, Toronto, Canada. Association for Computational Linguistics.

---

### Official Review · Reviewer_RsWP · 2023-11-01

**Soundness:** 3 good
**Presentation:** 3 good
**Contribution:** 3 good
**Rating:** 6
**Confidence:** 4

**Summary:**

This paper proposes a framework to combine Large Language Models (LLMs) and Specialized Translation Models (STMs). Specifically, The authors first sample translation candidate from a STM. Then they combine 1) source sentence, 2) STM's translation and 3) ground-truth reference into a triplet. By providing 10 triplets as demonstrations, LLMs learn through in-context learning and refine STM's translation. Further experiments show its superiority in low-resource translation and continual learning.

**Strengths:**

1. This work is comprehensive, with both the main experiment and the analysis experiment effectively conveying the author's perspective. The writing is clear and well-structured.

**Weaknesses:**

1. The idea is relatively straightforward. The SCALE framework and experiments are more engineering-oriented, lacking scientific insight.

2. The baselines used for comparison are weak. There are many previous works, such as back translation, pretraining, and other improvements for low-resource languages, which may require fewer resources and perform better.

3. Inference cost. The SCALE model involves two types of decoding:  STM decoding and LLM decoding. It is important to examine the computational cost associated with the inference process of the SCALE model.

**Questions:**

1. Why not conduct translation experiments on languages with slightly higher resources, such as German, French, or Chinese? In these languages, the improvements of SCALE may be limited.

---

> ### Author Response · Authors · 2023-11-19
>
> Dear Reviewer,
>
> Firstly, we would like to express our sincere gratitude for your time and effort in reviewing our paper. We appreciate your thoughtful feedback and are eager to address your concerns and questions.
>
> Concern #1: Lack of scientific insight
>
> Response: We acknowledge your viewpoint, but we stand firmly to dispute your claim that our work is predominantly engineering-oriented and lacks scientific insight. We wish to assert with utmost certainty that our work is not confined to the realm of engineering but makes significant strides in contributing to the scientific literature in these crucial ways:
>
> 1. Originality: We have pioneered an approach that seamlessly integrates two distinct translation paradigms within a single framework. This is a first in the field and enriches the existing body of knowledge by demonstrating how these two asymmetric translation models can work in synergy through the mechanism of triplet in-context learning.
> 2. Rigorous Experimentation: Our research encompasses stringent experiments that authenticate our approach. These tests extend beyond mere engineering prowess to involve scientific investigation, presenting empirical proof of how triplet in-context learning can bolster the performance of translation models in a variety of scenarios, including refinement, pivoting, and updating.
> 3. In-depth Analysis: We have conducted a thorough scrutiny of the translation traits of STM, LLM, and SCALE, delineating the subtle dissimilarities among these three systems. Furthermore, we have undertaken a sweeping analysis of SCALE to validate each critical design in our framework, enhancing our comprehension of SCALE in terms of its robustness and latency cost.
> 4. Far-reaching Implications: The outcomes of our research have profound implications for the design of future translation systems. They equip practitioners with strategic insights on when to employ large language models, specialized models, or a blend of the two. Additionally, our study tackles a pivotal question in machine translation literature in the era of Large Language Models: Should we continue to train specialized translation models, or should we wholeheartedly adopt LLM-based translation? Our paper breaks new ground by proposing a feasible solution that amalgamates these two systems within a single framework.
>
> We are confident that this articulates the scientific merit of our work. We will make it a point to highlight these aspects more prominently in our revised manuscript to forestall any potential misconceptions.
>
> ---
> Concern #2: Weak baseline
>
> Response: We would like to clarify that we have selected three types of baselines for our study: supervised models (NLLB, M2M100), few-shot LLMs (XGLM, GPT-3.5, and GPT-4), and one commercial translation system (Microsoft Translator). To our knowledge, these models are among the strongest and most representative of their respective categories. In particular, NLLB and MS Translator have already employed techniques such as back-translation and pretraining, and NLLB remains the state-of-the-art model on the Flores datasets.
>
> Moreover, our SCALE framework is not restricted to certain translation models. We believe that SCALE would also benefit from other strong baseline models by employing it as STM. If you could provide any additional models you believe we overlooked, we would greatly appreciate it. This would allow us to further validate and enhance the applicability of our SCALE framework.
>
> ---
> Concern #3: Inference cost.
>
> Response: We appreciate your valid concerns regarding the computational cost associated with the inference process of the SCALE model.
>
> In Section 5.4 of our paper, we have indeed addressed this concern. We have considered two additional costs brought about by SCALE: the STM decoding and the extended context for LLM inference. Our conclusion is that the latency incurred can be primarily attributed to the extended context window due to the transformer's quadratic time complexity. For more details, we kindly refer you back to our paper.

---

> ### Author Response · Authors · 2023-11-19
> **Response to  Reviewer RsWP (continued)**
>
> Question: Experiments on high-resource languages
>
> Response: In our experiment design, we primarily evaluate SCALE in the challenging low-resource setting. This decision was informed by current research findings that indicate Large Language Models already perform comparably with supervised models on high-resource languages, yet struggle with low-resource languages \[1][2]. Therefore, we believe it is more meaningful and challenging to focus on improving LLMs in this lower-resource setting.
>
> To resolve some of your concerns, we still would like to test SCALE performance in the high-resource setting. We carried out experiments in three languages on the Flores datasets: German, Czech, and Chinese, under the same settings as SCALE-refine. We utilized COMET-22 and COMETKiwi to measure the translations. As shown below, the results reveal that SCALE offers consistent improvements even for these high-resource languages, albeit less significantly than for low-resource languages as you predicted, and it exhibits robust performance even when paired with a less performant STM (especially in en-zh direction). This is remarkable, given the already strong performances of models such as GPT-4 and NLLB. All the generated output has been updated to the supplementary zip files for your reference.
>
> We assure you that these insights will be integrated into the revised version of our manuscript to provide a more comprehensive understanding.
>
> |                | en-de         | de-en         | en-cs         | cs-en         | en-zh         | zh-en         |
> | -------------- | ------------- | ------------- | ------------- | ------------- | ------------- | ------------- |
> | few-shot GPT-4 | 88.5/84.4     | **90.0**/84.6 | 92.0/86.8     | 88.4/85.0     | 88.8/**84.7** | 87.3/84.3     |
> | NLLB-3.3B      | 86.9/82.9     | 89.1/84.2     | 90.1/84.8     | 88.4/85.5     | 78.0/70.9     | 86.1/83.7     |
> | SCALE          | **89.0/85.1** | 89.9/**84.7** | **92.4/87.1** | **89.2/86.0** | **89.1/84.7** | **87.8/84.9** |
>
> We hope that our responses have addressed your concerns adequately. We would greatly appreciate it if you could consider revising your score in light of these clarifications. Please feel free to ask any further questions or voice any other concerns you might have.
>
> [1] Garcia, Xavier, et al. "The unreasonable effectiveness of few-shot learning for machine translation." *International Conference on Machine Learning*. PMLR, 2023.
>
> [2] Vilar, David, et al. "Prompting palm for translation: Assessing strategies and performance." *arXiv preprint arXiv:2211.09102* (2022).

---

### Author Response · Authors · 2023-11-20

Dear Reviewers,

I hope this message finds you well. As the end of the rebuttal phase is fast approaching (Nov 22 AOE), I wanted to kindly request your feedback on the amendments and clarifications provided in the rebuttal. Your insights and critiques are extremely important to me, and they greatly contribute to enhancing the quality of my work.

I understand the multitude of responsibilities you have, and I genuinely appreciate the time and effort you dedicate to this review process. Therefore, I kindly ask if you could possibly find the time to review the rebuttal and update your score accordingly if you find it appropriate.

I am more than willing to provide further clarifications or engage in additional discussions if needed.

Thank you once again for your time and consideration.

---

### Meta-Review · Area_Chair_XFw8 · 2023-12-16

**Metareview:**

This paper proposes a simple technique for MT where Specialized Translation Models (STMs) provide candidates which are refined by general-purpose Large Language Models (LLMs). Experiments show gains with respect to both few-shot LLMs (GPT-4) and specialized models (NLLB) in challenging low-resource settings. The reviewers pointed out some weaknesses, such as exclusive focus on low-resource languages and on XX-En direction only, as well as some surprising findings which seem to require further investigation (such as the quality of the selected demonstrations not affecting the final result). The authors addressed some of these concerns in the rebuttal phase, by reporting experiments in high-resource languages and in both directions. However, some concerns still remain regarding the need for a deeper analysis and investigation of questions that were left open. Furthermore, the new experiments considerably changed the flavour of the paper, which needs to be positioned differently in the face of the new experiments. I believe the paper would benefit from a rewrite consolidating this new information and a fresh round of reviews.

**Justification For Why Not Higher Score:**

Some concerns still remain regarding the need for a deeper analysis and investigation of questions that were left open. Furthermore, the new experiments considerably changed the flavour of the paper, which needs to be positioned differently in the face of the new experiments. I believe the paper would benefit from a rewrite consolidating this new information and a fresh round of reviews.

**Justification For Why Not Lower Score:**

N/A

---

### Decision · Program_Chairs · 2024-01-16

Reject